# Determination of Body Fat Ratio Standards in Children at Early School Age Using Bioelectric Impedance

**DOI:** 10.3390/medicina56120641

**Published:** 2020-11-25

**Authors:** Petr Kutac, Václav Bunc, Martin Sigmund

**Affiliations:** 1Human Motion Diagnostics Center, University of Ostrava, 701 03 Ostrava, Czech Republic; 2Faculty of Physical Education and Sport, Charles University, Praha 6, 162 52 Praha, Czech Republic; bunc@ftvs.cuni.cz; 3Application Centre BALUO, Faculty of Physical Culture, Palacký University, 771 47 Olomouc, Czech Republic; martin.sigmund@upol.cz

**Keywords:** body mass index, body fat, growth chart, prepubescent children, standards

## Abstract

*Background and objectives*: Body mass index (BMI) is commonly used to assess the proportionality of body mass; however, there are currently no standards for assessing the weight status of the child population for the needs of epidemiological studies. This study aims to establish bioelectric impedance analysis (BIA) standards for assessing the body weight of children (body fat, visceral fat) using BMI percentile growth charts. *Materials and Methods*: The study was implemented in a group of 1674 children (816 boys and 858 girls), ages 6 to 11. To classify the subjects at a percentile level, the percentile growth charts from the 6th national anthropological study in the Czech Republic were used. Body composition parameters were ascertained by BIA. *Results*: Body fat (%) and visceral fat standard values were determined for all age categories. The standards were in three-stages, enabling the determination of underweight, normal weight and overweight children aged 6–11 years. For boys with proportionate body mass, standard body fat values ranging from 14.3–16.0% to 15.5–18.0% were determined, while for girls’ values ranging from 16.7–19.4% to 18.3–20.5% were determined, depending on age. As far as visceral fat is concerned, standard values in boys ranging from 30.3–36.9 cm^2^ to 36.1–44.9 cm^2^ and in girls 30.3–36.9 cm^2^ to 36.1–44.9 cm^2^ were determined, depending on age. *Conclusions*: Standards for assessing weight status are applicable to children aged 6–11 years, while it can be confirmed that BMI can be considered as an objective tool in assessing body mass and body composition in children.

## 1. Introduction

Obesity is a considerable global problem in contemporary society. In advanced countries, its prevalence in adults as well as children is rapidly rising [1]. The transfer of obesity into adulthood is a significant risk of childhood obesity [2,3]. Obese children in adulthood may suffer from cardiovascular diseases, metabolic syndrome, diabetes mellitus Type 2, joint and other diseases with obesity as a risk factor. Childhood obesity can thus increase the morbidity in adulthood and thus reduce the quality of life [4]. To successfully influence it, corrective measures have to be implemented as soon as possible. Relevant diagnostics are a fundamental precondition. Therefore, it is important to pay attention to body weight, its fractioning and above all, the influence of body weight early in childhood; the assessment of body mass is still a frequently discussed issue [5]. A proportional body mass and its control are also considered to be indicators of a healthy lifestyle [6,7,8]. The relationship between body height and body mass, which can be used for the assessment of current body mass, can be studied using a series of weight–height indexes [9]. Body mass index (BMI) is frequently used for assessing the weight status of both children and adults in epidemiological studies. This index has been recommended since 1985 for the assessment of an appropriate body mass [10] and it is used for defining overweight or obesity [11] based on the results of extensive population studies in the USA. However, its use, especially in children, has proved problematic because it was widely used when combined with percentile growth charts. In this case, the observed individual’s BMI assessment is based on his/her position in the percentile band growth charts. When assessing adolescent children, it is essential to proceed from their biological age, which has to be respected when using growth charts. The World Health Organisation (WHO) also uses BMI in assessing proportionate body mass and thus the state of health of individuals [12], and constructs BMI development curves—BMI dependence on age. At the same time, WHO also formulates standards for assessing the development of body mass in children [13]. In the Czech Republic, there have been some national measurements of children and youth where BMI was used for assessments of the state of health and development of the population. The last such measurement was the sixth national anthropological study of the Czech Republic (6th NAS) in 2001 [14]. Only some anthropometric dimensions were measured during this national study (body height, body mass, circumferential dimensions, leg length and width). However, no parameters that could be used for assessments of body fat mass were measured. Considering the fact that the weight–height indexes, and therefore BMI, cannot assess the fractionation of body mass, some authors claim that their conclusive evidence, particularly in assessing overweight and obesity, is insufficient [15,16].

To assess the health and physical condition of an individual, it is recommended to examine both subcutaneous fat and visceral fat. It has been documented that an increased ratio of body fat mass is a risk factor in the occurrence of cardiovascular diseases [17]. At the same time, it is possible to use body fat mass values to identify metabolic syndrome with moderate accuracy [18]. Increased visceral fat, which is more active metabolically, is also considered to be a risk factor in cardiovascular diseases and obesity. [19,20,21]. The determination of body composition can be a suitable method for assessments of the current state of health and lifestyle of adults and children. This is documented by studies that deal with the effect of appropriate physical activity on body composition [22,23,24,25,26]. A variety of methods can be used to determine body composition in practice. Prediction equations represent the basic problem of indirect methods for determining body composition, as they are dependent on population, age, sex, body height, body mass and the estimated amount of body fat mass. The issue lies in the fact that various prediction equations often produce diametrically different results and also in their low validity against referential methods. A study [27] examined the inter-relationship of body composition, derived from simple anthropometry (BMI and skinfolds), bioelectric impedance analysis (BIA) and dual-energy X-ray absorptiometry (DXA) in children at the age from 3 to 8. The results showed that all the methods significantly underestimated body fatness as determined by DXA, and, overall, the various methods and prediction equations are not interchangeable.

The most frequently used referential methods, from laboratory methods, for the assessment of body composition in vivo are DXA, computerised tomography (CT) and magnetic resonance imagining (MRI). However, these methods are demanding to implement and also expose study participants to radiation (CT, DXA) [28]. Considering the fact that it is difficult to obtain consent for non-medical exposure of children and youth in many countries (including the Czech Republic), the BIA method, which is fast and simple, is frequently used, especially in epidemiological studies of those age groups [28]. To obtain the values of body composition parameters, prediction equations included in the BIA device’s software are used. Since each manufacturer uses their own equations, these prediction equations are often the source of very different results, even though it is still the BIA method, which has also been confirmed in a study by Kutáč and Kopecký [29]. In this study, the participants were measured by four different BIA analysers (tetrapolar single-frequency, tetrapolar multi-frequency 20–100 kHz, tetrapolar multi-frequency 1–1000 kHz and bipolar single-frequency hand–hand) consecutively. The differences found indicate that it is very complicated to create standards for body fat mass representation, unless uniform hardware and software are used to provide a sufficient accuracy in diagnosing overweight or obesity. In practice, it would mean the use of the same BIA analyser, as well as the same prediction equations.

Whatever applies to the method of all-body BIA analysis, it also applies to other methods of determining body composition that are used in practice. This does not only concern field methods (e.g., BIA and anthropometric) but also laboratory methods, such as DXA, MRI, CT, air displacement plethysmography (Bod Pod) or near infrared interactance (NIRI). This is probably why BMI is still widely used in epidemiological studies. Determining the BIA analyser and population-specific standards for classifying weight status are warranted for measuring and tracking body mass abnormalities. For practical use, the most suitable are the three-stage standards that classify individuals by weight status as underweight, healthy and overweight to obese, which are specific for each BIA analyser and population.

There are studies that deal with the validity of the use of BMI [30,31] or inverted BMI (iBMI) [30] to estimate %body fat mass (BF) in children and youth. However, these studies use two skinfolds for the estimate of %BF mass (subscapularis and triceps); the results indicate a mutual linear dependence and the estimate is used to determine a low or a high risk for being overweight or obese. Both of these studies show a linear relationship between BMI and %fat, which was determined by measurement of two skinfolds (subscapularis and triceps). In both of these studies, the relationships were used to estimate a low or high risk of overweight or obesity. A working hypothesis can be defined based on the aforementioned studies: it is possible to determine the weight status of the paediatric population based on the relationship between BMI development charts and body composition assessment determined by whole-body bioimpedance methods and whether this relationship is dependent on the weight status of the monitored children.

The main aim of this study is to assess the probative value of the BMI growth charts for assessing the weight status of children between the ages of 6 and 11. The second aim is to provide the values of body fat and visceral fat for the percentile bands of *p* < 25, *p* 25–75 and *p* > 75 in both sexes in the determined age group, measured by the BIA method and valid for the Czech child population.

## 2. Methods

### 2.1. Participants

The research group included 1674 participants (816 boys and 858 girls) aged 6 to 11 years old. They were children from the Moravian Region were selected in a non-random manner (Moravian-Silesian and Olomouc regions). Children were selected from cooperating elementary schools. There are 282,630 children at the age from 0 to 14 y living in the region [32]. Unfortunately, there are no population records in the age range we recorded (6–11 years); thus our number of 1674 children represents 0.59% of all children at the age of 0–14 in the region. Considering that our study only recorded six age groups and not 14, our sample most likely represents more than 1% of the total population in the determined age group of the region. These were prepubescent children, except for 15 11-year-old girls, who had reached menstruation. These girls were not excluded from the measurement, despite already having menstruated, with regard to their chronological age. As far as boys were concerned, we assessed their development (the occurrence of puberty) according to the methodology by Mirwald et al. [33] and Müller et al. [34]. We classified the recorded participants by chronological age. The count in the individual age categories is presented in Table 1. Classification into the corresponding age category was done according to the WHO classification, where individuals are classified into age categories after crossing the chronological age in an annual range (e.g., 6 years old = 6.00–6.99 years old) [35]. All the recorded children were Caucasian race (White race). The criteria of the study included the children’s medical condition (children without any objective medical complications on the basis of a medical check by a paediatrician), chronological age (6–11) and absence of regularly organised physical activity realised in sports clubs. The objective was to prevent the participation of children with specific sports training. The physical activity of the recorded children was random and it did not exceed 60 min per day. Children had compulsory physical education at school twice per week, with each session lasting 45 min. The decision to start at 6 years of age was based on the possibility of contacting the children in the long-term for the needs of intervention programmes for potential body mass reduction or fitness improvement, which is only possible during school attendance. All the participants participated voluntarily in the research, they were informed about the measurement procedure in advance and their parents or legal representatives signed an informed consent to participate in the study. The research was approved by the Ethics Committee of Palacký University on 28 November 2016 under file No. 76/2016 and is in compliance with the Helsinki Declaration. Neither the participants nor their legal guardians received any incentives to ensure their participation in the study. The study was implemented within a non-profit event.

### 2.2. Procedures

The measurements were taken in the medical rooms of each school, which are designated for treating potential injuries of students (especially during physical education (PE) classes). The measurement was done as a cross-sectional survey and conducted once for each participant. The measurement was implemented by the same research team under the supervision of the first author. The assessment of body fat mass and other anthropometric parameters took place in the morning, with observance of all the principles of measurement stated in the professional study dealing with the issues of the use of the BIA method [36]. Because BIA analysers measure body fluid volumes as a basic parameter, and the current state of hydration of the body depends on fluid intake, it is necessary to control their intake in the period before measurement when using these methods. Because the children studied were minors, the parents received accurate instructions that the children had to follow prior to measurement. These include the absence of intense physical activity 24 h prior to measurement, no food and drink 4 h prior to measurement and absence of caffeine 12 h prior to measurement. These recommendations are based on the recommendations in the aforementioned study [36]. The children’s guardians also received individual recommendations for drinking prior to measurement to ensure corresponding hydration in the measured children, based on the children’s body mass, physical activity and season. The actual fulfilment of these recommendations was checked by questioning just before the start of using the BIA analyser. The children were measured in sports clothing (shorts and a T-shirt) and barefoot.

The parameters measured were body height (BH), total body mass (BM), total representation of body fat (BF) mass and visceral fat area (VFA) expressed by area (cm^2^). BH was measured using a Tanita HR 001 stadiometer (Tanita, Japan). BM and BF mass were measured by a tetrapolar bioimpedance multi-frequency InBody 770 analyser (Biospace, South Korea) according to the recommendations of the manufacturers. The InBody 770 analyser was used in this study due to its worldwide use in diagnostic practice. The presented values of the monitored body composition parameters were not calculated but they were obtained from the InBody 770 BIA analyser.

The 6th NAS growth charts [14] were used for classification in the BMI percentile bands; the graphs are presented in Figure 1 and Figure 2. BMI assessment according to percentiles: P_1_ < 25, thin; P_2_ 25–75, proportionate; P_3_ > 75, plump, overweight or obese. First, we measured the BH and BM of each participant and calculated BMI using these parameters. Based on the resulting BMI and the current chronological age, the participant was registered in the corresponding percentile range of the 6th NAS growth chart, which is the current Czech population standard [14].
(1)$$\mathrm{BMI}\;=\;\frac{\mathrm{BM}\;\left(\mathrm{kg}\right)}{{\mathrm{BH}}^{2}\left(m\right)}$$

### 2.3. Data Analysis

The normality of distribution was verified by the Shapiro–Wilk test. The parametric independent *t*-test was used for assessment of the statistical significance of the differences in the mean of fat representation between neighbouring percentile bands. The level of statistical significance for all the tests was set at α = 0.05. Practical significance was assessed using the effect size (ES) of Cohen (Cohen’s *d*). A *d* value at the level of 0.2 indicates a minor change, 0.5 an intermediate change and 0.8 a major change [37]. A value of Cohen’s *d* ≥ 0.5 is considered practically significant. The practical significance between the groups, independent of the sample size, was assessed using the Cohen effect size.

The results were statistically processed using IBM SPSS Statistics (Version 21 for Windows; IBM, Armonk, NY, USA).

## 3. Results

Table 2 and Table 3 present the prevalence of participants within each percentile band in the individual age categories, the mean values of the recorded somatic parameters and standard values for the representation of body fat mass and visceral fat in individual age categories.

Table 4 presents the results of verifying the inclusion of the recorded participants in the percentile bands. All the recorded parameters had a normal distribution. Therefore, the parametric t-test was used to assess the statistical significance.

Overweight and obesity, which is represented by the P_3_ percentile band, averaged 35.1 ± 7.3% (range across age categories: 21.8% to 40.2%) in boys and 27.1 ± 11.2% in girls (22.8% to 31.6%).

In each percentile band of the relevant age categories, we determined the mean value as well as the lower and upper confidence interval value for BF mass and VFA by direct measurements using BIA. We consider the BF mass and VFA values in the percentile band P_2_ that ranged within the lower and upper confidence interval as the standard for the respective age category. The VFA values for 6-, 7- and 8-year old boys are the exception, as they overlap between the percentile bands P_1_ and P_2_. This is due to the absence of statistical as well as practical significance between these percentile bands (Table 4).

In addition to the participants who were overweight and obese, there were also participants who were underweight in our set (<25th percentile). The mean prevalence of underweight in the group of boys was 16.3 ± 2.4%, and it was 22.3 ± 4.2% in girls.

Statistically and practically significant differences in all percentile bands were determined in the percentage of body fat mass representation in boys. The value of Cohen’s *d* was 0.8 and higher. No statistical or practical significance was demonstrated in VFA in 6-, 7- and 8-year-olds between the percentile bands of P_1_ and P_2_ (*d <* 0.5).

In girls, statistically and practically significant differences were ascertained in BF mass and VFA in all age categories among all percentile bands. In 6-, 7-, 9- and 10-year old girls, intermediate change (*d* ≥ 0.5) was ascertained between the percentile bands P_1_ and P_2_, while in all other cases, major change (*d ≥* 0.8) was determined. The results demonstrate the correctness of the inclusion of the recorded participants in the individual percentile bands.

## 4. Discussion

The study had two main aims, the first being to provide sex-specific values for BF mass in individual percentile bands and other BIA-derived adiposity measures for Czech children aged 6–11 years. The values found have potential benefit in clinical and wellness settings and in the prevention of childhood obesity. Individuals are frequently scanned by BIA machines, the use of which is convenient thanks to their relative simplicity, provided that population-adequate equations are used and the hydration of the organism is known. The second aim of this study was to identify the BIA and anthropometric measurements that are most capable of screening individuals with high levels of body fat mass.

We assessed the prepubescent development period when the development of BH, BM and thus BMI should be approximately linear. However, the various levels of maturation had already manifested in this period in some individuals. As a child’s age increases, there is a gradual non-linear increase in BH, which is related to an increase in BM as well. This is natural ontogenetic development [38,39,40] and although the body mass increase is not proportional, it is related to the biological maturation of each individual, especially in puberty. This is also reflected by BMI changes related to age, which are not linear; therefore, the increase in BMI is also not linear [41].

The differences in the development of BH between children in P_2_ and P_3_ are insignificant (Figure 3A,C). Boys in P_3_ have insignificantly higher BH than boys in P_2_ by 1.7% to 3.3%; girls in P_3_ have insignificantly higher BH than girls in P_2_ by 0.7% to 3.3%. However, the differences in the development of BM between children in P_2_ and P_3_ are significant (Figure 3B,D). Boys in P_3_ have higher BH than boys in P_2_ by 21.6% to 25.7% and girls in P_3_ have higher BM than girls in P_2_ by 21.5% to 27.2%. These findings also reflect the BMI values and indicate an inadequate increase in BM of the current child population.

Our values of the prevalence of overweight and obesity correspond with the WHO values, where the overweight and obesity prevalence range is very broad and thus it is not possible to cover such national dissimilarities [42,43].

The results of the comparison of BF mass and VFA values between adjacent percentile bands showed the absence of a significant difference only in the VFA values in 6-year-old, 7-year-old and 8-year-old boys between P_1_ and P_2_ (Table 4). This is likely caused by the higher sensitivity of boys with low body mass to changes in lifestyle (physical activity or nutritional habits) [44]. The statistically insignificant differences in boys could be the result of the low number of participants in the individual percentile bands, as well as the inhomogeneity of the recorded groups.

BIA-derived body composition measures had higher correlations with %BF mass than surrogate anthropometric measures. When analysing the results, it is essential to respect population dependence, adherence to the conditions of hydration of the organism and possible differences between the chronological and biological age. Especially during puberty, the assessment of biological age should be considered when interpreting the results and this age should be used in prediction equations. When evaluating body composition using BIA, it is preferable to use analysers with a tetrapolar electrode configuration. The position of the participant when being measured represents another important factor. A standing position, which is used with the InBody and Tanita analysers, is particularly suitable for the analysis of healthy individuals and for fast measurements in large groups of participants with lower demands on measurement accuracy. Often, a stable position, mainly in the case of individuals with neurological issues, seniors or young children, may be the problem. For clinical analyses, it is preferable to use analysers that measure a participant in a lying down position. However, their disadvantage is measurement asymmetry, where the electrodes are attached to only one half of the body, thus not ensuring the symmetrical measurement of the whole body. In practice, we do not come across individuals with the same halves of the body very often. Our results on the efficacy of the BMI tool in assessing weight status and body composition, through an examination of the body fat mass ratio, correspond with the results of a similar study [45]. Morimoto et al. [45] observed the relationship between the values of body fat mass measured via the BIA and BMI methods in two groups of children (9–10 years and 12–13 years). A strong (*r* = 0.78) and very strong (*r =* 0.98) relationship was noted between the determined values of the BF mass and BMI. The authors [45] also confirmed the possibility of using BMI to screen body mass in children in the field. The assessment of adiposity with the use of BMI and %BF with BIA (Tanita BC-418MA) in obese children from the age of 6.9 to 18.9 was implemented by Hunt et al. [46]. It demonstrated that there was a significant linear dependency between BMI and %BF and confirmed the use of BMI to determine adiposity in children. The aforementioned studies and our data imply that the relationship between BMI and directly measured %BF is independent of the weight condition of children and thus it is possible to use BMI to assess adiposity and weight status in Czech children.

The aforementioned studies and the data we have collected enable us to accept the set working hypothesis.

## 5. Limitations of the Study

There are two major limitations of this study. The first major limitation arises from the sample size. When the participants were distributed by gender and age into the individual percentile bands (three percentile bands), the number of participants in the bands was low. It should be noted that all somatic parameters were directly measured and that they were not determined based on a questionnaire survey. Therefore, the range of the sample can be considered relevant.

The second limitation concerns the applicability of the obtained results. With regard to the fact that we used the BMI growth charts of 6th NAS Czech Republic, prepared on the basis of measurement implemented in 2001, whereas our measurement was implemented in 2017, this time difference may be significant and the graphs would then fail to fully reflect the current facts. These graphs were used because our participants were Czech children and there are no more recent growth charts available in the Czech Republic. However, we believe that our results indicated, despite these limitations, that the growth charts with the percentile bands can be successfully used in the field to assess overweight or obesity in the Czech child population.

## 6. Conclusions

It was manifested that the values of BM and BF mass representation increase with the increasing percentile band. The differences found between percentile bands were significant. The study results indicate that the BMI development percentile graphs and thus BMI are valid for body mass screening in the field, not only for BIA methods but also for the calliper method [28,29] or other methods of determining body composition. The study thus confirms the possibility of using percentile growth charts for the assessment of overweight and obesity using BMI in children at early school age. When the percentile growth chart is up-to-date, BMI may be considered as an objective tool for field assessment of BM, body composition and lifestyle in prepubescent children.

The study established values of body fat mass and visceral fat, determined on the basis of a representative sample of the child population in Czechia (6th NAS), for proportionate body mass in the 6–11 age categories, which are valid when using the tetrapolar multi-frequency BIA analyser InBody 770. The findings apply to the population of prepubescent Czech children.

## Figures and Tables

**Figure 1 medicina-56-00641-f001:**
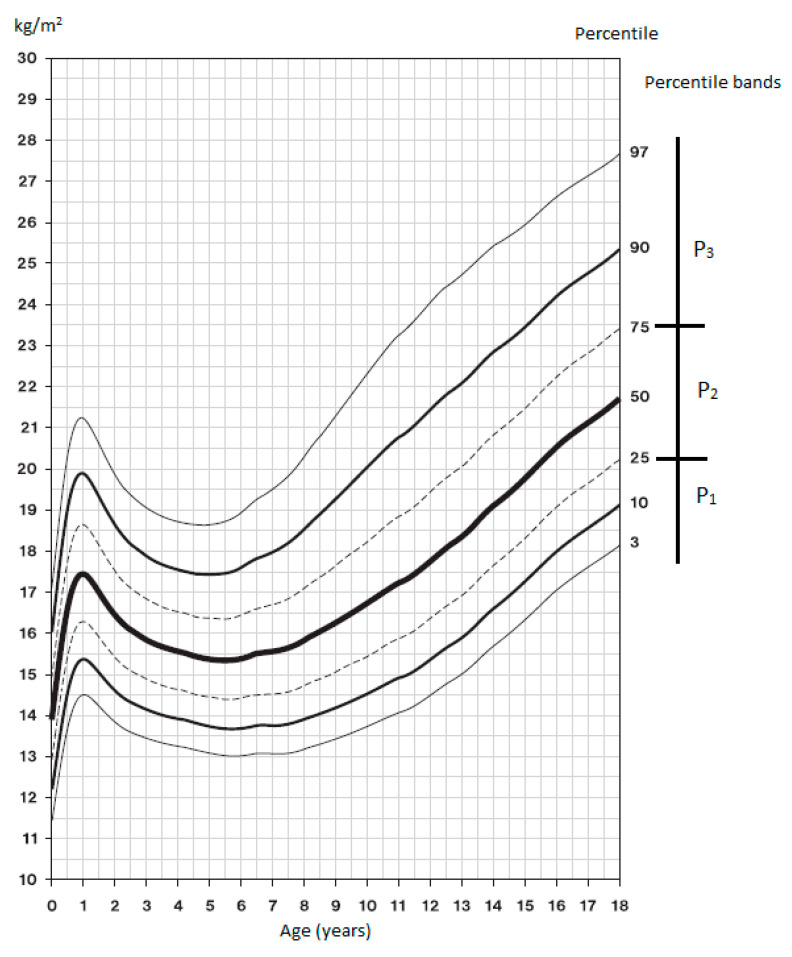
BMI percentile growth chart for boys—6th NAS [14].

**Figure 2 medicina-56-00641-f002:**
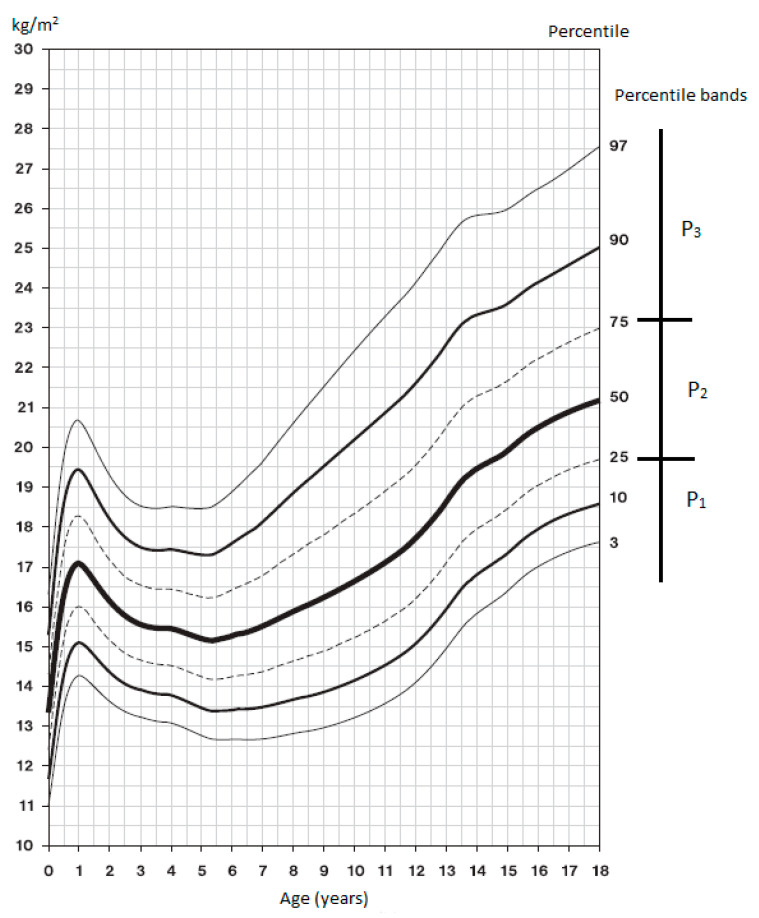
BMI percentile growth chart for girls—6th NAS [14].

**Figure 3 medicina-56-00641-f003:**
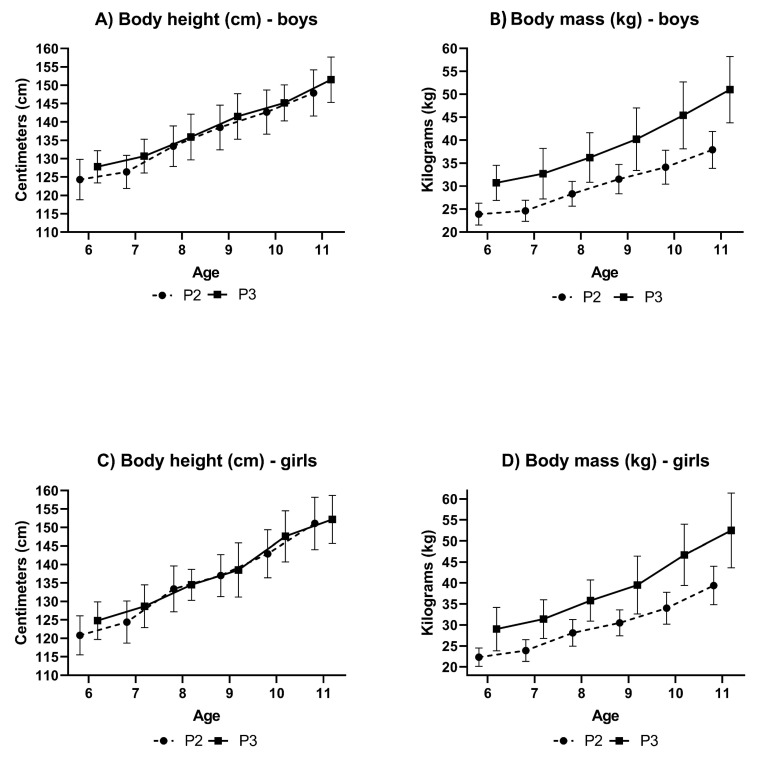
Development of body height and body mass in children in percentile bands P_2_ and P_3_.

**Table 1 medicina-56-00641-t001:** Participants in the measured age categories.

	6 Years	7 Years	8 Years	9 Years	10 Years	11 Years
Boys	110	138	152	149	159	108
Girls	87	165	168	171	156	111

**Table 2 medicina-56-00641-t002:** Mean values of the somatic characteristics of body fat mass (BF) and visceral fat (VFA)—boys.

Age (years)	P	*n* (%)	BH (cm) M ± SD	BM (kg) M ± SD	BF (%) M ± SD (95% CI)	VFA (cm^2^) M ± SD (95% CI)
6	P_1_ < 25	22 (20.0)	123.1 ± 4.9	20.9 ± 1.9	11.6 ± 2.1 (10.6, 12.5)	33.7 ± 8.7 (29.8, 37.6)
	P_2_ 25–75	64 (58.2)	124.3 ± 5.5	23.9 ± 2.4	15.2 ± 3.8 (14.3, 16.1)	36.6 ± 12.4 (34.0, 40.0)
	P_3_ > 75	24 (21.8)	127.8 ± 4.4	30.7 ± 3.8	25.0 ± 5.8 (22.6, 27.5)	53.8 ± 15.2 (47.4, 60.2)
7	P_1_ < 25	20 (14.5)	126.3 ± 5.4	21.9 ± 2.1	10.8 ± 2.5 (9.6, 12.0)	30.7 ± 9.5 (26.1, 35.2)
	P_2_ 25–75	76 (55.1)	126.4 ± 4.5	24.6 ± 2.3	15.1 ± 3.6 (14.3, 16.0)	34.3 ± 12.1 (31.5, 37.1)
	P_3_ > 75	42 (30.4)	130.7 ± 4.6	32.7 ± 5.5	25.2 ± 7.6 (22.9, 27.6)	57.5 ± 20.4 (51.2, 63.9)
8	P_1_ < 25	20 (13.1)	132.5 ± 4.7	24.6 ± 2.1	11.7 ± 3.4 (10.1, 13.4)	30.1 ± 11.4 (24.7, 35.6)
	P_2_ 25–75	73 (48.0)	133.4 ± 5.5	28.3 ± 2.7	16.3 ± 4.1 (15.4, 17.3)	35.6 ± 11.8 (32.8, 38.4)
	P_3_ > 75	59 (38.9)	135.9 ± 6.2	36.2 ± 5.4	26.6 ± 7.8 (24.5, 28.6)	62.1 ± 21.4 (56.5, 67.7)
9	P_1_ < 25	22 (14.8)	134.6 ± 4.8	25.8 ± 1.9	12.1 ± 3.2 (10.7, 13.6)	24.9 ± 13.8 (18.6, 31.2)
	P_2_ 25–75	68 (45.6)	138.5 ± 6.1	31.5 ± 3.2	16.7 ± 3.3 (15.9, 17.5)	35.4 ± 10.9 (31.0, 37.5)
	P_3_ > 75	59 (39.6)	141.5 ± 6.2	40.2 ± 6.8	27.1 ± 7.0 (25.3, 28.9)	67.5 ± 21.9 (61.8, 73.2)
10	P_1_ < 25	27 (17.0)	142.7 ± 9.5	29.6 ± 4.1	10.7 ± 3.6 (9.3, 12.2)	20.4 ± 10.6 (16.1, 24.7)
	P_2_ 25–75	68 (42.8)	142.7 ± 6.0	34.1 ± 3.7	16.7 ± 5.2 (15.4, 18.0)	35.7 ± 14.7 (30.3, 36.9)
	P_3_ > 75	64 (40.2)	145.2 ± 4.9	45.4. ± 7.3	29.8 ± 6.5 (28.1, 31.4)	77.7 ± 25.1 (71.4, 83.9)
11	P_1_ < 25	20 (18.5)	147.8 ± 4.0	32.7 ± 1.9	13.2 ± 3.6 (11.5, 15.0)	23.7 ± 9.8 (19.0, 28.4)
	P_2_ 25–75	46 (42.6)	147.9 ± 6.3	37.9 ± 4.0	16.8 ± 4.2 (15.5, 18.0)	39.0 ± 14.2 (36.1, 44.9)
	P_3_ > 75	42 (38.9)	151.5 ± 6.2	51.0 ± 7.2	29.8 ± 7.5 (27.4, 32.1)	83.9 ± 27.9 (75.2, 92.6)

P, percentile; *n*, frequency; BH, body height; BM, body mass; BF, body fat mass; VFA, visceral fat area; M, mean; SD, standard deviation; 95% CI, confidence interval.

**Table 3 medicina-56-00641-t003:** Mean values of the somatic characteristics of body fat mass (BF) and visceral fat (VFA)—girls.

Age (years)	P	*n* (%)	BH (cm) M ± SD	BM (kg) M ± SD	BF (%) M ± SD (95% CI)	VFA (cm^2^) M ± SD (95% CI)
6	P_1_ < 25	24 (27.6)	120.3 ± 4.2	19.4 ± 1.9	12.1 ± 2.6 (11.0, 13.3)	28.7 ± 13.0 (23.1, 34.3)
	P_2_ 25–75	41 (47.1)	120.8 ± 5.3	22.3 ± 2.2	18.0 ± 4.3 (16.7, 19.4)	37.8 ± 13.4 (35.5, 42.1)
	P_3_ > 75	22 (25.3)	124.8 ± 5.1	29.0 ± 5.2	28.2 ± 5.3 (25.9, 30.6)	56.8 ± 14.2 (50.5, 63.1)
7	P_1_ < 25	39 (23.6)	124.5 ± 9.0	20.6 ± 2.1	12.7 ± 3.6 (11.5, 13.9)	26.5 ± 10.4 (23.1, 30.0)
	P_2_ 25–75	74 (44.8)	124.4 ± 5.7	23.9 ± 2.6	18.6 ± 4.0 (17.4, 19.3)	34.7 ± 13.1 (31.6, 37.8)
	P_3_ > 75	52 (31.6)	128.7 ± 5.8	31.4 ± 4.6	29.1 ± 6.2 (27.4, 30.8)	58.2 ± 17.9 (53.2, 63.2)
8	P_1_ < 25	44 (26.2)	129.5 ± 5.2	22.5 ± 2.2	11.3 ± 4.3 (9.9, 12.6)	20.5 ± 12.5 (16.6., 24.3)
	P_2_ 25–75	76 (45.2)	133.4 ± 6.2	28.1 ± 3.2	18.1 ± 3.8 (17.2, 19.0)	33.7 ± 13.8 (30.5, 36.8)
	P_3_ > 75	48 (28.6)	134.5 ± 4.2	35.8. ± 4.9	30.1 ± 6.2 (28.3, 31.9)	62.8 ± 17.1 (57.9, 67.8)
9	P_1_ < 25	35 (20.5)	136.8 ± 5.1	26.1 ± 2.0	13.9 ± 3.7 (12.6, 15.2)	23.7 ± 11.3 (19.8, 27.6)
	P_2_ 25–75	97 (56.7)	137.0 ± 5.7	30.5 ± 3.1	18.7 ± 4.3 (17.9, 19.6)	34.3 ± 16.1 (31.0, 37.5)
	P_3_ > 75	39 (22.8)	138.5 ± 7.3	39.5 ± 6.9	31.5 ± 6.3 (29.4, 33.5)	69.2 ± 20.0 (62.8, 75.7)
10	P_1_ < 25	24 (15.4)	140.5 ± 6.7	28.7 ± 2.7	15.1 ± 3.6 (13.5, 16.6)	23.0 ± 11.8 (17.9, 28.1)
	P_2_ 25–75	85 (54.5)	142.9 ± 6.5	34.0 ± 3.8	18.6 ± 4.8 (17.6, 19.7)	33.6 ± 15.2 (30.3, 36.9)
	P_3_ > 75	47 (30.1)	147.6 ± 6.9	46.7 ± 7.3	32.4 ± 6.7 (30.5, 34.4)	81.2 ± 21.8 (74.8, 87.6)
11	P_1_ < 25	22 (19.8)	145.2 ± 7.7	31.1 ± 3.4	13.4 ± 3.5 (11.8, 15.0)	19.9 ± 9.6 (15.5, 24.3)
	P_2_ 25–75	62 (55.9)	151.1 ± 7.1	39.4 ± 4.6	19.4 ± 4.3 (18.3, 20.5)	40.5 ± 17.2 (36.1, 44.9)
	P_3_ > 75	27 (24.3)	152.2 ± 6.5	52.5 ± 8.9	31.8 ± 7.2 (28.9, 34.6)	82.3 ± 28.8 (70.9, 93.7)

P, percentile; *n*, frequency; BH, body height; BM, body mass; BF, body fat mass; VFA, visceral fat area; M, mean; SD, standard deviation; 95% CI, confidence interval.

**Table 4 medicina-56-00641-t004:** Verification of the correctness of inclusion in percentile bands: differences in the mean values of body fat mass (BF) and visceral fat (VFA) between adjacent percentile bands.

Age (years)	P	Boys		Girls	
		Δ BF (%)	Δ VFA (cm^2^)	Δ BF (%)	Δ VFA (cm^2^)
6	P_1_ vs. P_2_	−3.6 ***	−2.9 ^NS^	−5.9 ***	−9.1 *
	P_2_ vs. P_3_	−9.8 ***	−16.6 ***	−10.2 ***	−19.0 ***
7	P_1_ vs. P_2_	−4.3 ***	−3.6 ^NS^	−5.9 ***	−8.2 **
	P_2_ vs. P_3_	−10.2 ***	−23.2 ***	−10.8 ***	−23.5 ***
8	P_1_ vs. P_2_	−4.6 ***	−5.5 ^NS^	−6.8 ***	−13.2 ***
	P_2_ vs. P_3_	−10.2 ***	−26.5 ***	−12.1 ***	−29.2 ***
9	P_1_ vs. P_2_	−4.6 ***	−10.5 **	−4.8 ***	−10.6 **
	P_2_ vs. P_3_	−10.4 ***	−32.2 ***	−12.7 ***	−35.0 ***
10	P_1_ vs. P_2_	−6.0 ***	−15.3 ***	−3.5 **	−10.6 **
	P_2_ vs. P_3_	−13.1 ***	−42.0 ***	−13.8 ***	−47.7 ***
11	P_1_ vs. P_2_	−3.6 **	−15.3 ***	−6.0 ***	−20.6 ***
	P_2_ vs. P_3_	−13.0 ***	−44.8 ***	−12.3 ***	−41.8 ***

P, percentile; BF, body fat mass; VFA, visceral fat area Δ difference; * *p* < 0.05; ** *p* < 0.01; *** *p* < 0.001, ^NS^ not significant.

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
