# Peer review of "Determination of Body Fat Ratio Standards in Children at Early School Age Using Bioelectric Impedance"

_medicina, 2020, doi:10.3390/medicina56120641_

Round 1
Reviewer 1 Report
I struggle with the focus of this manuscript. I see the true novelty being that large-scale anthropometric assessments of Czech youth have not been conducted for nearly two decades (see lines 18-21). As such, a very useful outcome of this work could be to create updated Czech-specific BMI percentile charts for 6-11-year-old girls and boys. Likewise, what are the take-home messages for clinicians? For example, body fat percentages greater than XX in boys and YY in girls are cause for concern.
I also struggle with the generalizability of this research. In addition to occurring in the Czech Republic, excluding children who have health conditions and/or participate in sports eliminates many.
The references should be updated for focus more on children. When possible, recent references should be incorporated.
Comments:
- Line 7: Use a reference specific to children. The current citation focuses on middle age.
- Lines 13-14: Why are the percentile growth charts problematic in children? Is this more problematic than BMI assessment in adults? If so, why?
- Lines 32-33: What does “body fat (BF) distribution” mean in the sentence? Circumference measurements? Skinfolds?
- Lines 36-38: This reference focused on ballet dancers. It is not appropriate to the focus population of this study.
- Line 42: The words “to radiation” should follow “expose study participants.”
- Line 58: What do “2x indirect” and “1x indirect” mean?
- Line 65: What is iBMI?
- Line 90: Were the pubescent girls excluded from the analysis? How was puberty assessed in males?
- Line 96: Is it correct that all children with health issues were excluded? For example, asthma and allergies are quite common among children.
- Line 107: Were any participation incentives provided?
- Line 110: Where did the assessments occur? For example, were they conducted at school?
- Line 128: How was VFA determined? Is this calculated by the InBody BIA analyzer?
Potentially helpful references:
- https://doi.org/10.2105/ajph.82.3.358
- https://doi.org/10.1016/j.amepre.2011.07.003
Reviewer 2 Report
The author took up an interesting topic, taking the common problems of overweight and obesity among children and youth into consideration. Nevertheless, the manuscript requires considerable corrections. In some places, there are significant ambiguities, intricacies and the usual stylistic errors which I discuss below:
Introduction
- Line 6 – this sentence needs notes.
- Line 7-8 and 11, 127 and Table 3 – the authors used the terminology: mass, body weight, body mass. Are these synonyms, according to the authors? Or these expressions might have different meanings? If so, it should be clarified at manuscript (e.g. using a glossary). The potential reader will be confused.
- Line 30 – Why the authors wrote [13, 14-17]? It would be better [13-17].
- Line 68 – „” or obese Both of”. Probably there is lack of a dot.
- Line 72 and line 74 – compare: „of BF” and „of BF mass”. What is the difference between these terms (semantic)?
- Line 71 - the deletion was left.
- Line 71-78 - the objectives of the research have been written intricately. Furthermore, the authors used different terms like: “the aim”, “the goals” and “purposes” (line 194). Why? The scientific text should be unambiguous.
- Line 77-78 – the information about „analyser” in this part of manuscript is wrong. It should be put in part 2 (Methods).
Methods
- Line 84 and 89 – compare “1,674” and “1674”. Why were different records used?
- Line 80-87 – Study setting - I strongly doubt that this description can be considered as a Study setting. This description rather shows how the selection of people for the study group proceeded.
Results
- Line 156 – It was written “Table 2 and 3 present …”. It should be “In the table 2 and 3 were presented…”.
- Table 2 – age 9 – there is lack of brackets. It should be (15.9, 17.5).
Discussion
- Line 226 – The authors wrote „As hypotheisized…”. In which part of manuscript (line) did you write hypothesis?
- In the discussion, the authors refer only to their own research. In the discussion also the results of own research with the research of others (similar research) should be also compared. This part of the discussion is missing.
References
- It is necessary to double-check and unify the entries and enter the access date in the case of items from electronic sources. Please check: item 2, 4, 11, 23, 26, 30.
Round 2
Reviewer 1 Report
The manuscript has been improved but still insufficient.
Regarding the comment about the appropriateness of the ballet dancer reference, the purpose of this study was to validate body composition assessment of a specific population. The median age of participants was 19 and all were female; the focus of this study is boys and girls ages 6 to 11. The ballet dancer reference focuses on low body weight athletes trained in classical ballet; in this study, children who regularly engaged in organized physical activity were excluded.
Author Response
Regarding the comment about the appropriateness of the ballet dancer reference, the purpose of this study was to validate body composition assessment of a specific population. The median age of participants was 19 and all were female; the focus of this study is boys and girls ages 6 to 11. The ballet dancer reference focuses on low body weight athletes trained in classical ballet; in this study, children who regularly engaged in organized physical activity were excluded.
We removed the reference of Leal et al. (2019) – ballet dancers in the text and replaced it with a new one [27] that deals with the inter-relationship of body composition derived from simple anthropometry (BMI and skinfolds), BIA and DXA in children at the age from 3 to 8. We also modified the part of the text related to that reference.
We also modified other parts of the text, namely the discussion and the conclusion, according to the recommendations of the second reviewer.
Reviewer 2 Report
Dear Authors,
thank you for answering my questions and correction the manuscript.
In my opinion, the manuscript correction is still insufficient.
Instead of supplementation of data (eg. line 6), authors removed the sentence. Authors didn't answer directly for a question (line 226). The discussion was not completed enough.
Apart from, the studies carried out do not justify the use of the terms "normative reference value" or "body ratio standards". Authors did not present reliability for the developed data.
Author Response
Instead of supplementation of data (eg. line 6), authors removed the sentence.
We have added new information to the text, including references.
Authors didn't answer directly for a question (line 226). The discussion was not completed enough.
We added the hypothesis – in the Introduction. We added information about the acceptance of the hypothesis in the Discussion.
Apart from, the studies carried out do not justify the use of the terms "normative reference value" or "body ratio standards". Authors did not present reliability for the developed data.
When categorising the children we monitored into the corresponding growth graph bands for BMI, we used the Czech normative data (CAV). We do not consider the data we obtained to be “normative” with respect to the reviewer’s comment, which we agree with, however, considering the scope of the sample we measured, we can consider the data to be very important for the needs of the Czech clinical and wellness practice and for the prevention of child obesity. We modified the text.
Round 3
Reviewer 1 Report
Thank you for responding to the feedback of the reviewers. This manuscript has been improved.
Reviewer 2 Report
No comments and suggestions.